# Design and development of an irrigation monitoring and control system based on blynk internet of things and thingspeak

**Fahmy Rinanda Saputri** [1]*, **Ricardo Linelson**[1], **Muhammad Salehuddin**[1], **Danial Md Nor**[2], **Muhammad Imran Ahmad** [3,4]

1 Department of Engineering Physics, Universitas Multimedia Nusantara, Tangerang, Indonesia,
2 Department of Electronic Engineering, Universiti Tun Hussein Onn Malaysia, Batu Pahat, Johor, Malaysia, 3 Institute of Sustainable Agrotechnology (INSAT), Universiti Malaysia Perlis, Arau, Perlis, Malaysia, 4 Faculty of Electronic Engineering & Technology, Universiti Malaysia Perlis, Arau, Perlis, Malaysia

* fahmy.rinanda@umn.ac.id

## Abstract

The scarcity of water resources exacerbated by climate change poses a major challenge for sustainable agriculture. This study presents an Internet of Things (IoT)-enabled irrigation system designed for real-time monitoring and precise water control. Using the Blynk platform and ThingSpeak for data management, the system integrates sensors for soil moisture, temperature, and humidity with a NodeMCU module to optimize irrigation practices. Initial results demonstrate the system's effectiveness in improving water use efficiency and supporting sustainable agricultural practices, providing a low-cost, accessible solution for small and medium-scale farmers.

## 1 Introduction

The increasing extremity of global climate change has created significant challenges for the agricultural sector, particularly in managing limited water resources [1,2]. Rising temperatures and shifts in soil moisture patterns disrupt traditional irrigation methods, posing risks to crop yields. In response, Internet of Things (IoT)-based automatic irrigation systems have emerged as a promising solution, enabling real-time, demand-based watering guided by environmental data from sensors [3–6]. These systems enhance water use efficiency by utilizing sensors to monitor soil moisture, temperature, and other environmental factors that affect crop water requirements, providing smart solutions to address climate change challenges [7]. This study stands out by integrating Blynk for real-time control and ThingSpeak for cloud data storage, offering a more effective, accessible, and cost-efficient solution for farmers facing the impacts of climate change.

Previous research, such as the development of an IoT-based smart agriculture framework that integrates renewable energy, precision irrigation, and Android app-based control, as well as the implementation of a real-time fire detection system to improve the resilience of agricultural operations, demonstrate progress in utilizing smart technology for sustainability [8–10]. In addition, previous studies also have advanced IoT-based irrigation systems to enhance water use efficiency. For instance, Hari et al. developed an IoT irrigation system

**Data availability statement:** All relevant data are within the manuscript and its Supporting Information files.

**Funding:** The author(s) received no specific funding for this work.

**Competing interests:** The authors have declared that no competing interests exist.

**List of abbreviations:** Arduino IDE: Arduino Integrated Development Environment; DC: Direct Current; DHT11: Digital Humidity and Temperature 11; IoT: Internet of Things; LCD: Liquid Crystal Display; NodeMCU V3: Node Microcontroller Unit Version 3; PLC: Programmable Logic Controller; SDGs: Sustainable Development Goals; UV-C: Ultraviolet-C; Wi-Fi: Wireless Fidelity

utilizing a Programmable Logic Controller (PLC), though this system lacked cloud-based data storage for archiving historical data [11]. Reddy et al. incorporated cloud technology in their IoT irrigation system, yet did not focus on sensor accuracy and precision [12]. Additionally, studies by Stolojescu-Crisan et al. and Velmurugan et al. emphasized soil moisture sensors and weather prediction but did not integrate ThingSpeak for data storage and Blynk for simultaneous control and monitoring [13,14]. This study addresses these gaps by integrating Blynk for real-time irrigation control and ThingSpeak for cloud-based data storage, enabling both immediate feedback and long-term insights. This combination provides farmers with enhanced control and the ability to analyze historical data to optimize irrigation practices and improve water efficiency, overcoming the limitations of previous systems.

This research addresses these limitations by incorporating water-resistant temperature sensors and capacitive soil moisture sensors (V2.0), along with rigorous testing for accuracy, precision, bias, and error, to improve data reliability. The IoT platform Blynk is used for real-time control, while ThingSpeak serves as a cloud database, creating an adaptable and implementable solution for farmers [15,16]. Additionally, the Node Microcontroller Unit Version 3 (NodeMCU V3) ESP-12 microcontroller, with Wireless Fidelity (Wi-Fi) connectivity and low power consumption, offers a cost-effective solution for small- to medium-scale farmers. This system integrates a Digital Humidity and Temperature 11 (DHT11) sensor for air humidity, a water-resistant DS1820 sensor for temperature, and soil moisture sensors to monitor environmental conditions in real-time. Irrigation frequency is calibrated to maintain optimal soil moisture at 30–50% and temperatures of 20–30°C, which are crucial for crop productivity.

The importance of real-time monitoring has been proven in various studies, such as those that monitor environmental conditions in buildings using various sensors connected via IoT [17,18]. Furthermore, this research will also design a real-time monitoring system for automatic irrigation, which enables efficient water management by utilizing soil moisture sensors and an IoT platform. Meanwhile, controlling the irrigation system can be done using the Blynk application, as applied in previous research which was used to control Ultraviolet-C (UV-C) lamps automatically via IoT [19]. This integration of IoT technologies in agriculture aims to improve water usage efficiency and support decision-making processes for farmers by providing them with real-time, data-driven insights into soil moisture levels.

The aims of this study are to evaluate the effectiveness of an automatic irrigation system in maintaining optimal soil moisture levels through sensor data, assess the impact of IoT technology on irrigation control and farmer decision-making, and analyze trends in soil moisture data to inform future irrigation practices. This approach aims to enhance water use efficiency in agriculture and support sustainable resource management in the context of climate challenges. Developed using Blynk for control and ThingSpeak for cloud storage, this system is implemented through Arduino Integrated Development Environment (Arduino IDE), allowing for seamless integration between hardware and software. Besides remote monitoring, the system features a 16x2 Liquid Crystal Display (LCD) for local display, providing real-time feedback to users on soil moisture levels. Additionally, a Direct Current (DC) mini pump is activated automatically to irrigate crops when soil moisture falls below the preset threshold, ensuring crops receive adequate water. Soil moisture sensor calibration has been performed to increase accuracy, ensuring reliable data for informed irrigation decisions and reducing water waste. [20]. By integrating Blynk IoT and ThingSpeak, this study provides a low-cost, accessible automatic irrigation solution particularly suited for developing regions with limited access to advanced technology. This innovation aligns with the United Nations Sustainable Development Goals (SDGs), specifically SDG 2: Zero Hunger, SDG 6: Clean Water and Sanitation, and SDG 13: Climate Action. By improving water use efficiency and promoting sustainable agriculture practices, it not only enhances resilience to climate change but also supports long-term

food security. This research contributes to addressing the impacts of climate change, ensuring sustainable agricultural development, and improving access to clean water resources, ultimately fostering a more resilient and sustainable future.

## 2  Materials and methods

This research was conducted to develop an automated irrigation monitoring and control system integrated with IoT-based data transmission to improve water usage efficiency and enable remote monitoring for agricultural applications. The research methodology is organized into several stages, including research design, system development and algorithm implementation, testing, and data acquisition. Each stage is explained chronologically below.

### 2.1  Research approach

This research employs a mixed method approach, combining quantitative and qualitative methodologies. The quantitative approach involves real-time data collection from sensors, including soil moisture, temperature, and air humidity, to evaluate the system's effectiveness in maintaining optimal land conditions. The qualitative approach is based on field observations and experiments conducted by researchers to assess the system's usability, flexibility, and practicality. This integrated approach ensures a comprehensive understanding of the system's performance in real-world scenarios.

### 2.2  Research procedure and algorithm

The research procedure involved both hardware and software development, combining a C++-based program and circuit assembly. The selection of the C++ language is based on the use of the Arduino IDE, which provides built-in libraries to support the integration of hardware such as sensors and relay modules. C++ enables efficient program development and makes it easy to test directly on the microcontroller, making it ideal for IoT applications.

The research procedure is organized into several phases to ensure the system operates effectively and meets the design specifications. The main stages include System Design, Programming and IoT Integration, and System Testing. In the System Design phase, the hardware and software components necessary for the system are identified. This includes wiring soil moisture sensors, temperature sensors, and relay modules to the NodeMCU, followed by developing a schematic diagram to visualize the connections between components. The Programming and IoT Integration phase involves writing the control program for the NodeMCU using the Arduino IDE. Additionally, the NodeMCU is integrated with the Blynk application and ThingSpeak platform, enabling real-time monitoring of soil moisture, temperature, and humidity data. Finally, in the System Testing phase, multiple tests are conducted to verify system performance. Sensor Testing ensures accurate readings of soil moisture, temperature, and humidity, while Relay Module Testing validates that the relay operates according to the control logic.

Wi-Fi and Cloud Connection Testing confirms stable data transmission to both Blynk and ThingSpeak, and Pump Control Testing evaluates the automated irrigation functionality by verifying that the pump activates when soil moisture drops below 60% and deactivates when the optimal threshold is met. This structured approach allows for the step-by-step validation of system functionality and reliability. The steps are as follows:

- System Initialization. During the initialization stage, the ESP8266 microcontroller configures various sensors and prepares the system for continuous data collection and analysis. The components include the DS18B20 temperature sensor, which monitors air temperature,

and the DHT11 humidity sensor, which measures air humidity. A capacitive soil moisture sensor is also initialized to assess soil moisture levels and determine irrigation needs. Additionally, a relay module is set up to control the irrigation pump, operating based on the readings from the soil moisture sensor. The relay module is chosen to control irrigation pumps because it provides a reliable and efficient way to interface low-power microcontrollers, such as NodeMCU, with high-power devices like water pumps. It acts as an electrically isolated switch, ensuring safe operation by preventing direct current surges from reaching the microcontroller.

- Data Acquisition and Processing. The system's sensors capture environmental data, including air temperature, air humidity, and soil moisture levels. This data is processed directly in the microcontroller using specific threshold values to classify soil moisture conditions. The system determines whether the soil moisture is below or above 60%. The 60% moisture threshold is chosen because it optimizes microbial activity, ensuring healthy soil respiration. At this level, the soil maintains a balanced water-air ratio, supporting efficient oxygen diffusion and nutrient absorption [21]. If the soil moisture level is below this threshold, irrigation is initiated; if it is at or above this threshold, irrigation stops. This straightforward, rule-based approach provides an efficient way to manage irrigation without complex algorithms

- Control Algorithm. The control algorithm employs conditional statements in C++ to automate irrigation. Based on soil moisture levels, the system activates or deactivates the pump as needed. The algorithm uses an if statement to check the soil moisture condition. The Control Algorithm for the automated irrigation system is embedded in the loop() function within the provided code, particularly under the Automatic Mode section.

- Automatic Mode (system == 1). The algorithm continuously monitors the soil moisture level (soilmoist). If the soil moisture is below the setpoint SP_HIGH (in this case, 60%) and the pump is not currently running (fp == 0), the pump is turned on (digitalWrite(pump, LOW);). The pump status fp is then set to 1 to indicate the pump is active. If the soil moisture reaches or exceeds the setpoint SP_LOW (in this case, 40%) and the pump is running (fp == 1), the pump is turned off (digitalWrite(pump, HIGH);). The pump status fp is reset to 0 to indicate the pump is inactive.

- Manual Mode (system == 0). When in manual mode, the pump's state is controlled by the user through the Blynk app, and the pump's state (ON or OFF) is displayed on the LCD screen. This structured approach provides a clear distinction between Automatic and Manual modes, ensuring flexible operation based on the user's preferences and environmental conditions. The algorithm is optimized for simplicity and efficiency, eliminating the need for complex logic such as fuzzy classification.

- LCD Display Output. The system displays sensor readings (temperature, humidity, soil moisture) on the LCD screen. It also shows the pump irrigation ON/OFF status based on the soil moisture level.

- Data Transmission to IoT Platforms. The system connects to Wi-Fi using the ESP8266's network capabilities and sends data to the Blynk IoT app and ThingSpeak cloud platform. Authentication tokens, SSID, and passwords are transmitted to establish secure connections. Once connected, the system uploads real-time sensor data to the IoT platforms, allowing remote monitoring

In the software part, this research uses the Arduino IDE as a programming platform to write, compile, and upload code to the NodeMCU. The Blynk app, which is IoT-based, enables

real-time monitoring and control of the system through a smartphone or PC. ThingSpeak is used as the IoT data storage platform, which serves to store and analyze long-term data from sensors, as well as provide visualization of soil moisture trends and environmental conditions. This combination of hardware and software supports automatic, real-time monitoring and control of irrigation.

## 2.3 Research design

The research began with a literature review to understand current automated irrigation systems, IoT integration for remote monitoring, and water conservation strategies. This review helped establish the system's requirements, which include real-time monitoring, automatic irrigation control based on soil moisture levels, and remote access to sensor data via IoT platforms such as Blynk IoT and ThingSpeak.

Based on these requirements, the system was designed to include multiple environmental sensors, a relay-controlled water pump, and an LCD 16x2 display for on-site monitoring. The ESP8266 microcontroller was selected as the main control unit for its Wi-Fi capability, enabling easy integration with the IoT platforms. Fig 1 illustrates the schematic diagram of the complete system, designed to map the interconnections between components. Based on Fig 1, the automatic irrigation system in this study utilizes several key components, including NodeMCU as the main microcontroller with Wi-Fi connectivity to control the entire system and send data to the IoT platform. The sensors used include Capacitive Soil Moisture Sensor V2.0 to measure soil moisture, Temperature Waterproof Sensor DS18B20 to monitor the ambient temperature, and Humidity Sensor DHT11 to record air humidity. In addition, the system is equipped with a 5V relay module that controls the water pump, a 16x2 I2C LCD to display data locally, and a 5V/2A power supply as a power source. Additional components

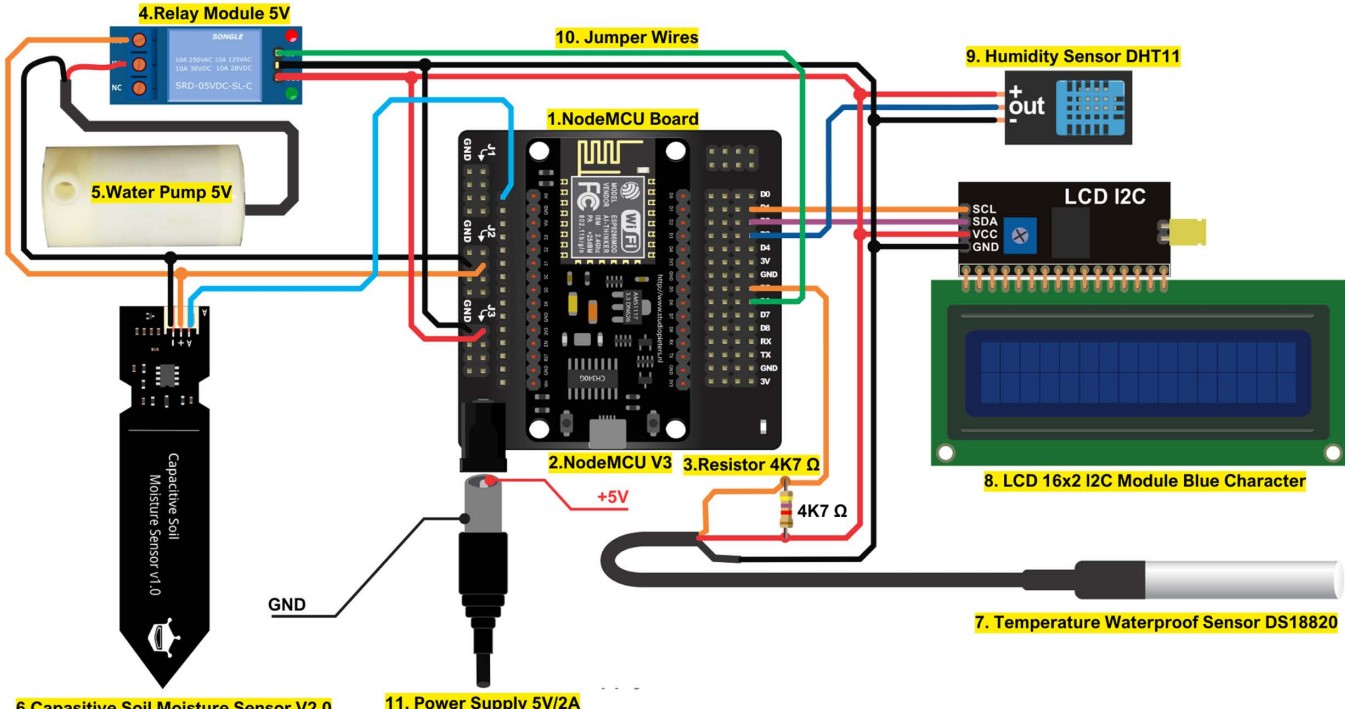

**Fig 1. Schematic diagram of the smart automatic irrigation monitoring and control system.**

such as a 4.7 kΩ resistor are used to support communication with the DS18B20 sensor, while jumper cables connect all components in the system.

This automatic irrigation system works by collecting real-time data from soil moisture, temperature (DS18B20), and air humidity (DHT11) sensors, which are processed by NodeMCU as the main controller module. Based on the soil moisture data, the NodeMCU regulates the activation of the relay module to turn on the water pump automatically if the moisture is below a predetermined threshold. The system is equipped with a 16x2 LCD screen for local monitoring and uses the Blynk platform for real-time control via mobile devices, as well as ThingSpeak for cloud-based data storage and long-term analysis. Table 1 summarizes the main hardware components used in the automatic irrigation system along with their respective functions. This table provides an overview of the components and their roles in the system.

The programming code for the NodeMCU is written using the Arduino IDE. The code includes setup for the NodeMCU to read data from soil moisture, temperature, and humidity sensors and to transmit this data to the Blynk app and ThingSpeak database. Additionally, it incorporates the logic for controlling the relay module, which manages the water pump status based on soil moisture levels. Finally, the NodeMCU configuration is completed to connect to a smartphone hotspot or Wi-Fi network.

The flowchart diagram of this automatic irrigation system encompasses three main aspects: the plant watering process, data transmission to the cloud database, and the data transmission and control flow of the system. This diagram illustrates the workflow for plant watering and data transmission to the cloud database, which includes Blynk and ThingSpeak. The Blynk application can be accessed via a smartphone app or through the website https://blynk.io/ from a smartphone, PC, or laptop. ThingSpeak can be accessed via https://thingspeak.mathworks.com/ using the same devices.

Fig 2 illustrates the process of the Automatic Irrigation Monitoring System, starting with the initialization of sensors, including the DS18B20 (temperature sensor), DHT11 (humidity sensor), soil moisture sensor, and relay. The system then detects sensor data, such as temperature, humidity, and soil moisture levels. This data is also sent in real-time to the Blynk app, which can be accessed via smartphone or PC, and stored in ThingSpeak for historical analysis. This process enables data access at any time and enhances data-driven irrigation management, making it easier to adjust based on monitored conditions.

Fig 3 shows if in Automatic Mode when the moisture is below 60%, the water pump is turned on; if it is above or equal to 60%, the water pump is turned off. If in Manual Mode, when the Pump Button is pressed ON, the pump will turn on. If the Pump Button is pressed OFF, the pump will turn off. The Pump Button can be pressed or controlled by using the Blynk IoT smartphone application or website. Table 2 presents a summary of the configuration parameters, system settings, and pump operation modes in the automatic irrigation

Table 1. Summary of hardware components and functions.

| Component | Function |
|---|---|
| NodeMCU (ESP8266) | Main microcontroller with Wi-Fi connectivity |
| Capacitive Soil Moisture Sensor V2.0 | Measures soil moisture |
| DS18B20 (Waterproof Sensor) | Measures ambient temperature |
| DHT11 | Measures air humidity |
| Relay 5V Module | Controls the water pump |
| LCD 16x2 I2C | Displays data locally |
| Power Supply 5V/2A | Provides power to the system |
| 4.7 kΩ Resistor | Supports communication with the DS18B20 sensor |

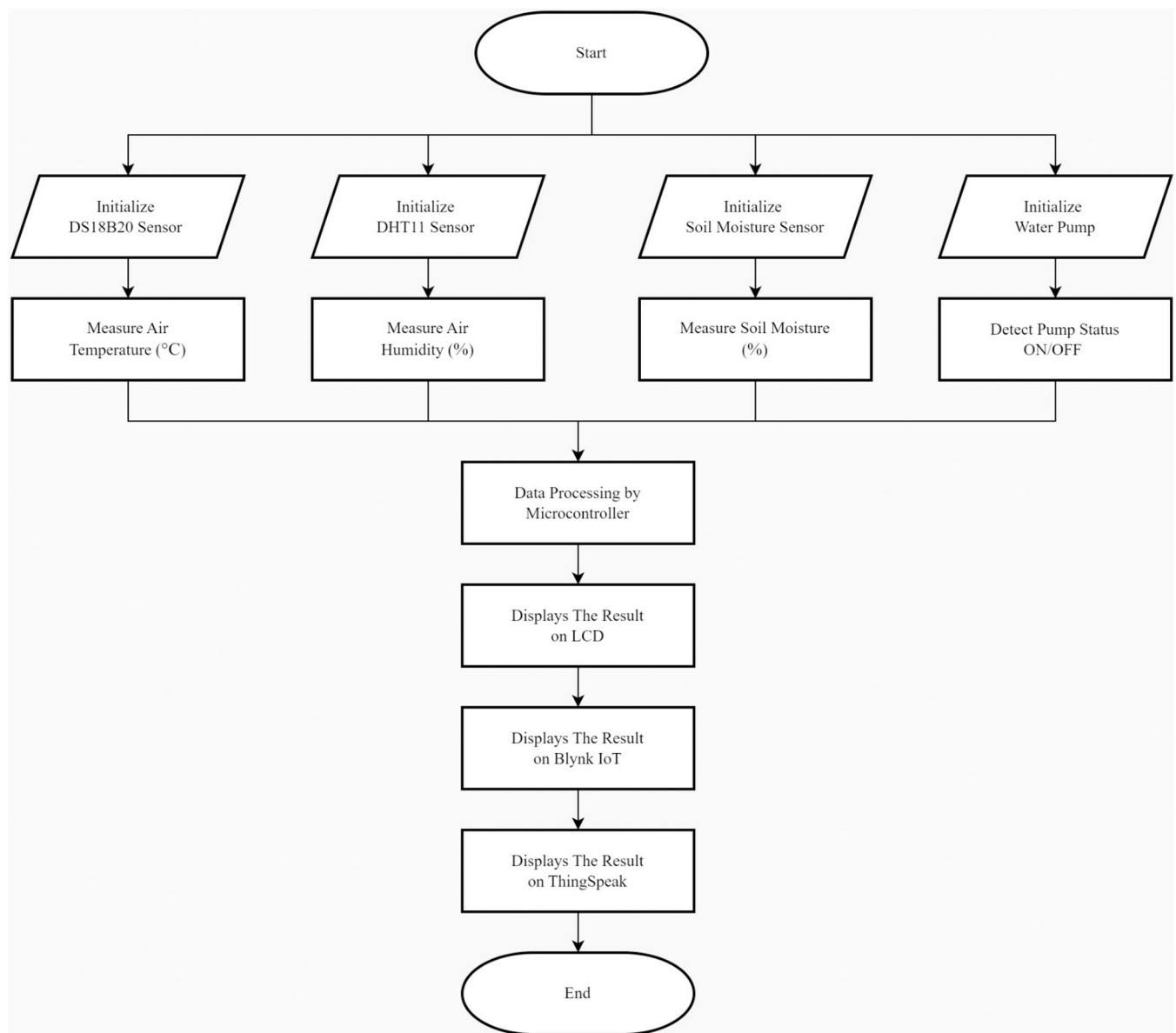

**Fig 2. Flowchart of the automatic irrigation monitoring system.**

system. These parameters define how the system operates under different conditions, both automatically and manually.

Block diagram in Fig 4 shows the process of data transmission and control in the IoT-based automatic irrigation system. In the IoT-based automatic irrigation system, input consists of data from soil moisture, temperature (DS18B20), and humidity (DHT11) sensors, which are sent to the NodeMCU for processing. The process involves the NodeMCU analyzing the sensor data to determine whether the soil moisture level is below a predefined threshold (e.g., 60%), in which case it activates the relay to turn on the water pump. If the soil moisture reaches the optimal threshold, the pump is automatically turned off.

The NodeMCU also connects to Wi-Fi to transmit the data to cloud platforms such as Blynk or ThingSpeak. The output includes real-time data displayed on the Blynk app, stored

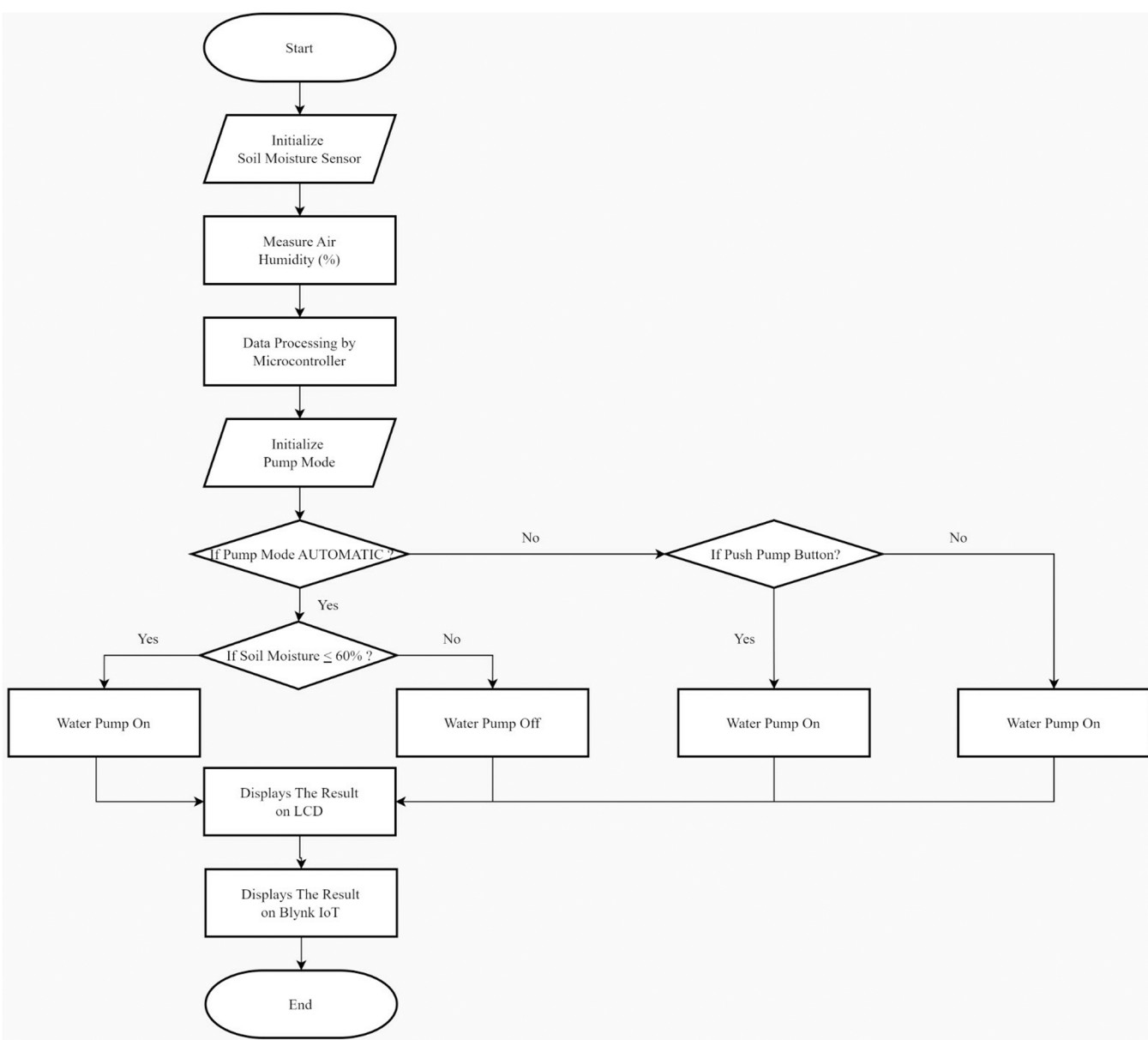

**Fig 3. Flowchart of the automatic irrigation control system.**

in ThingSpeak for long-term analysis, and shown on an LCD for local monitoring. This workflow enhances irrigation efficiency by enabling data-driven management and providing access to real-time and historical data via the cloud for improved decision-making. In addition, the system consumes 0.8 W when the pump is off and 1.0 W when the pump is on, making it energy-efficient for continuous operation.

## 2.4 Testing and data acquisition

The testing and data acquisition phase involved verifying the system's functionality and performance under controlled conditions. Functionality testing assessed the system's ability to automatically activate and deactivate the irrigation pump based on soil moisture levels, with

Table 2. Configuration parameters and pump operation modes.

| Parameter | Value/Description |
|---|---|
| Soil Moisture Threshold | Pump ON if < 60% |
| | Pump OFF if ≥ 60% |
| Operation Mode | Automatic: Based on sensor data |
| | Manual: via Blynk |
| IoT Platforms | Blynk (real-time control), ThingSpeak (data storage) |
| Wi-Fi Connection | Configured in NodeMCU with SSID & Password |
| Pump Operation (Automatic Mode) | < 60% Moisture → Pump ON |
| | ≥ 60% Moisture → Pump OFF |
| Pump Operation (Manual Mode) | Pump Button pressed ON → Pump ON |
| | Pump Button pressed OFF → Pump OFF |

controlled manipulation of soil moisture to observe corresponding pump responses. Accuracy and precision testing involved comparing sensor readings against reference measurements, with calibration adjustments made to ensure reliable data. Real-time data on soil moisture, temperature, and humidity were collected through Blynk and ThingSpeak, with historical data on ThingSpeak enabling analysis of soil moisture patterns. Sensor precision was calculated using the equation (1).

$$\mathrm{Pr}\,ecision = 100\% \left(1 - \frac{\sigma}{\bar{x}}\right) \tag{1}$$

where σ represents the standard deviation of the sensor readings and $\bar{x}$ is the mean of the sensor readings. The accuracy percentage of the sensors can be determined using Equation (2).

$$Accuracy = 100\% \left(1 - \frac{bias + 3\sigma}{x\,\mathrm{true}}\right) \tag{2}$$

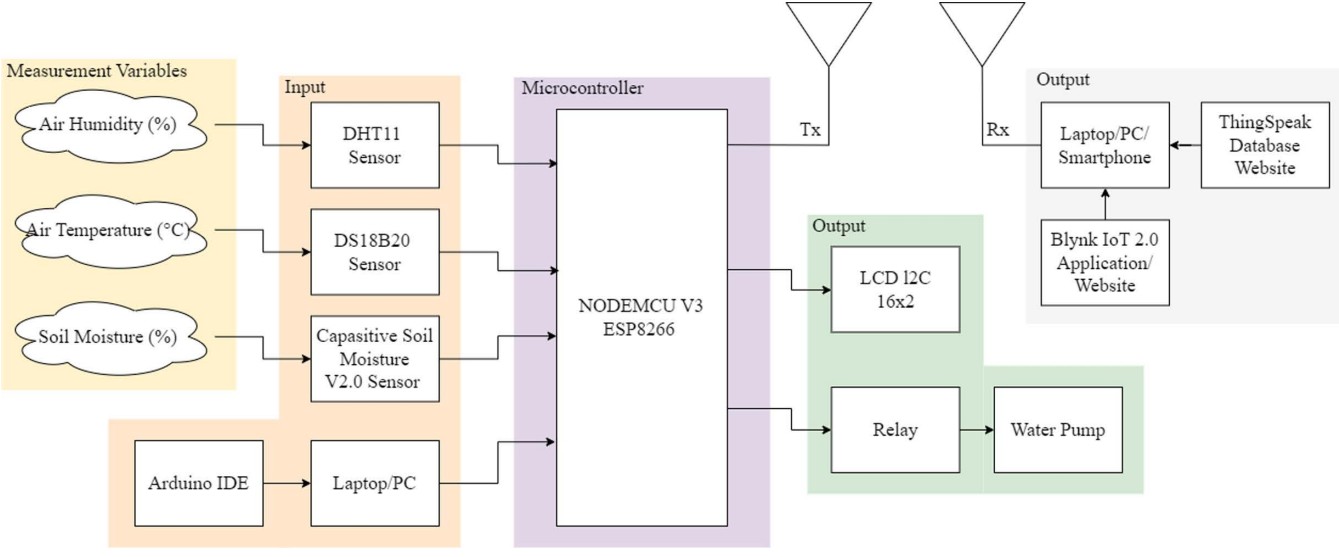

**Fig 4. Block diagram of irrigation and monitoring system.**

In this context, bias is calculated as the difference between the true value x true and the mean reading x̄, as shown in Equation (3). Then, to determine the percentage error of the sensors, Equation (4) is used.

$$Bias = \text{x true} - \overline{X} \tag{3}$$

$$error = 100\% \left( \frac{bias + 3\sigma}{x\ true} \right) \tag{4}$$

In these equations, σ represents the standard deviation of the sensor readings, x̄ is the mean of the sensor readings, and x true is the actual measured value by the calibrated tool. These calculations provide essential metrics for evaluating the sensor's precision, accuracy, bias, and error percentage.

IoT connectivity testing validated the Wi-Fi connection and successful data transmission to Blynk and ThingSpeak under various network conditions, ensuring reliable remote monitoring and data logging. Historical data logged on ThingSpeak supported environmental trend analysis and optimized automatic irrigation for efficient water usage.

The prototype of the IoT-based automatic irrigation system was tested outdoors in South Tangerang, Indonesia, which has tropical climate. South Tangerang has a tropical monsoon climate with high humidity and significant seasonal rainfall variations. The soil consists of a mixture of loam and sandy loam, which affects water retention and drainage. Challenges in the region include fluctuating weather conditions, high evapotranspiration rates, and heavy rainfall, all of which impact soil moisture and irrigation requirements [20]. The deployment aimed to validate system performance in real-world conditions, assessing its reliability in soil moisture monitoring, automatic watering, and IoT data transmission. The testing site was selected to ensure realistic conditions for evaluating water efficiency, system responsiveness, and data accuracy.

## 2.5 Ethical considerations

This study was conducted in accordance with the research contract approved by the Research and Community Service Institute (LPPM) of Universitas Multimedia Nusantara under contract Number 0007-RD-LPPM-UMN/P-INT/VI/2024. The research methodology and implementation adhered to the institutional guidelines and relevant regulations. No additional ethical approval was required as the study did not involve human or animal subjects. No deviations from the approved study protocol occurred during the research process.

## 3 Results and discussion

The results of research are provided in the figures and data visualization. Fig 5 illustrates the design of the automated irrigation system employed in this study. The researchers utilized an integrated Wi-Fi module with NodeMCU to ensure connectivity with the Blynk application, where the data is subsequently sent to the ThingSpeak database. For initial testing, the researchers employed a smartphone hotspot for convenience. However, in the final stage, the system will be fully integrated with a fixed Wi-Fi network at the farming site to enhance communication reliability and stability. This setup will ensure the system operates optimally for real-time irrigation monitoring and control.

The Blynk IoT dashboards such in Figs 6–8 provide an intuitive interface for monitoring and controlling the irrigation system. Initially, as shown in Figs 6–8 are the initial view, automatic control mode, and manual control mode respectively. In automatic control mode,

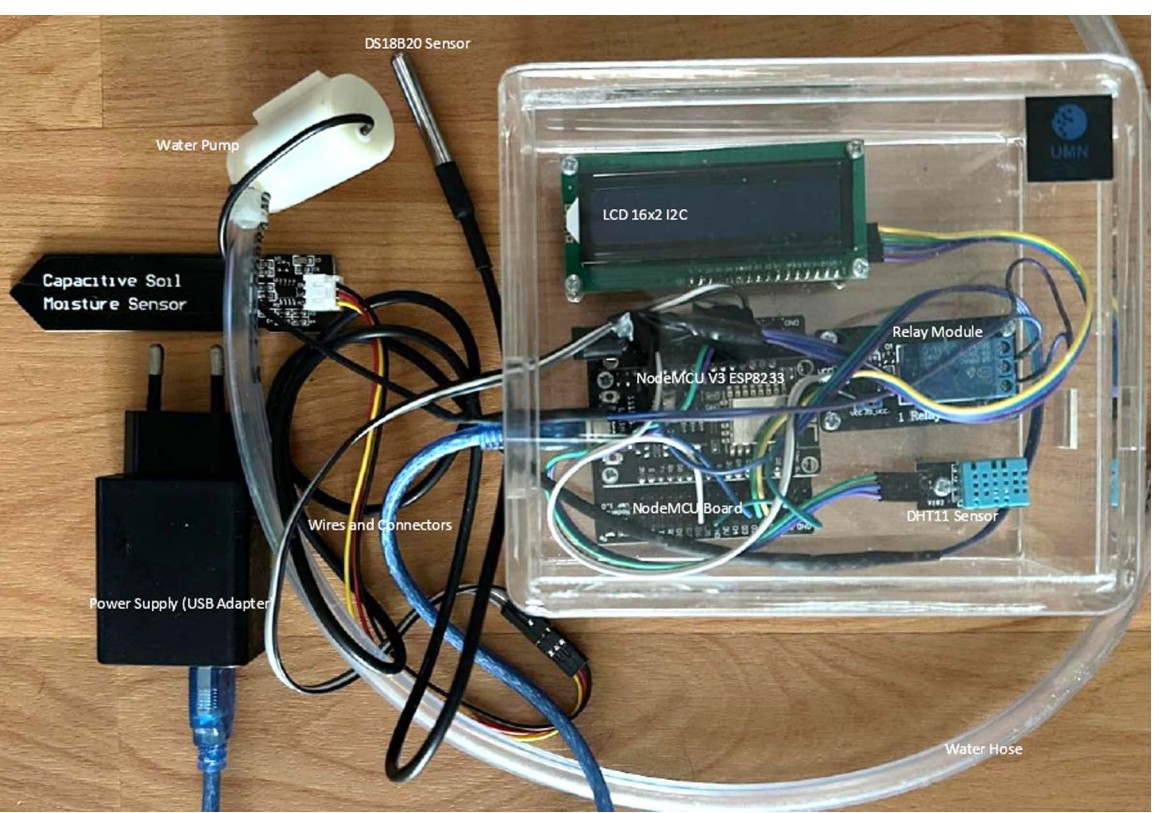

**Fig 5. Final circuit design.**

the pump will automatically turn on according to the preset threshold. In manual mode, the user can manually activate the pump by pressing the "Pump ON" button. Then, Fig 9 is the ThingSpeak Database Website Dashboard.

### 3.1 Bias, precision, accuracy, and error testing

This study conducted efficiency and accuracy tests to evaluate the performance of the DS18B20 and DHT11 sensors in measuring temperature and humidity. The measurements were compared against data from a hygrometer, which served as the calibration reference.

### 3.2 Air temperature measurement with DS18B20 sensor

Table 3 presents the air temperature measurements obtained using the hygrometer and the DS18B20 sensor, including bias, precision, accuracy, and error values. The average bias for the DS18B20 sensor was −0.9°C, with a bias of −0.8, precision of 98.4%, accuracy of 96.9%, and error of 3.1%. High precision and accuracy indicate that the DS18B20 sensor delivers highly accurate results, with minimal deviation from the hygrometer used as a reference.

Table 4 presents the results of the air temperature measurements using the hygrometer and DS18B20 sensor after calibration, including bias, precision, accuracy, and error. The average temperature measured by the DS18B20 sensor was 30.0°C with a standard deviation of 0, bias of 0.7°C, precision of 100%, accuracy of 97.8%, and error of 2.2%. The high precision and accuracy indicate that the DS18B20 sensor delivers excellent performance after calibration.

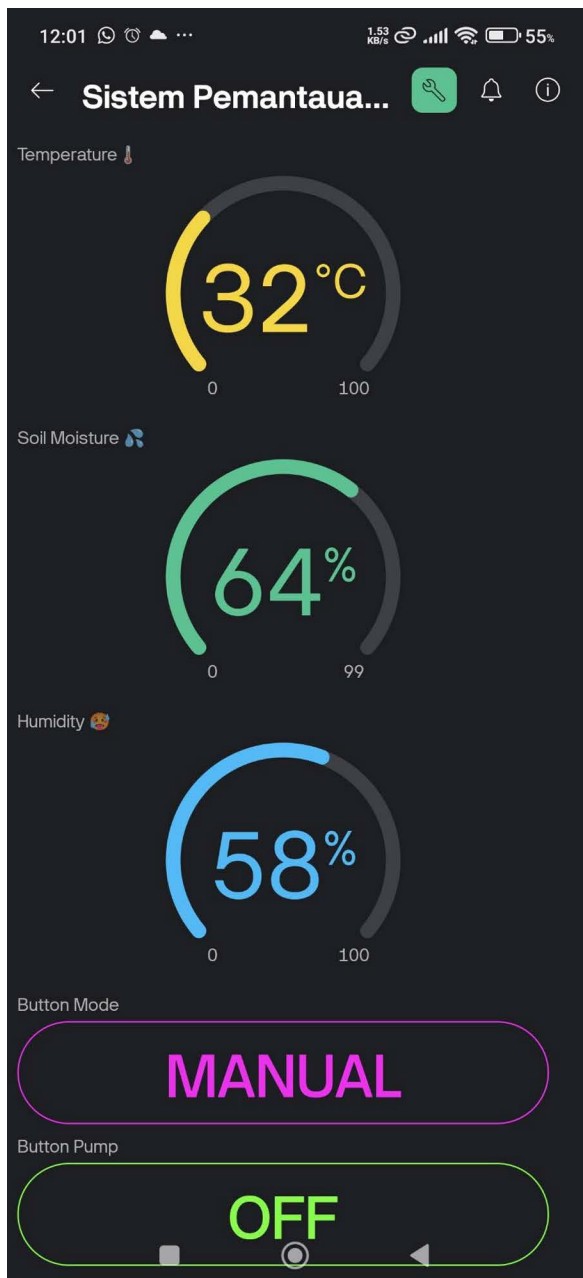

**Fig 6. Blynk IoT mobile app dashboard - Initial view.**

### 3.3 Air humidity measurement with DHT11 sensor

Table 5 displays the air humidity measurements obtained from the hygrometer and the DHT11 sensor, including bias, precision, accuracy, and error values. The DHT11 sensor exhibited an average bias of −0.2, precision of 98.8%, accuracy of 96.7%, and error of 3.3%. Table 6 shows the results of air humidity measurements obtained from the hygrometer and DHT11 sensor after calibration. The DHT11 sensor exhibited an average humidity of 65.0%, with a standard deviation of 0%, bias of 1.0%, precision of 100%, accuracy of 100%, and error of 0.8%. Despite the high precision and accuracy, minor variations with the hygrometer data

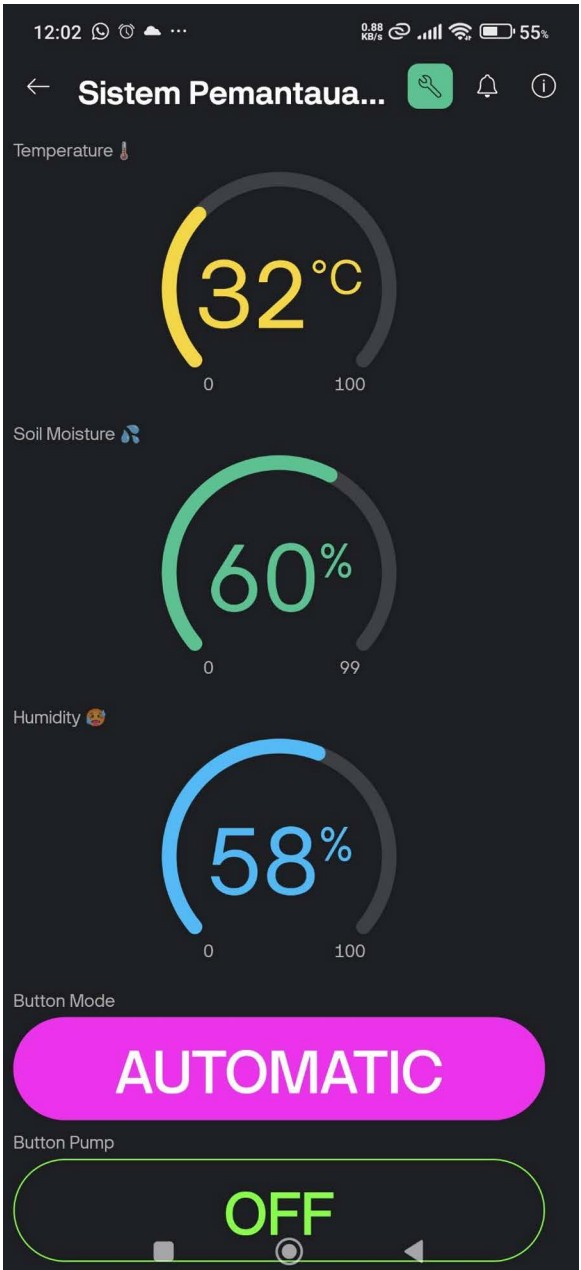

**Fig 7. Blynk IoT mobile app dashboard - Automatic control mode.**

were observed, indicating that the DHT11 sensor, while reliable, may require further monitoring. However, the study is limited by restricted field testing, and the system's long-term reliability remains unverified. Future research should focus on extended field trials to assess durability and performance across diverse agricultural environments and climates.

### 3.4 Soil moisture measurement with capacitive soil moisture sensor V2.0

Table 7 presents data showing the increase in soil moisture in response to the amount of water injected, measured in milliliters (mL). The data, recorded by the Capacitive Soil

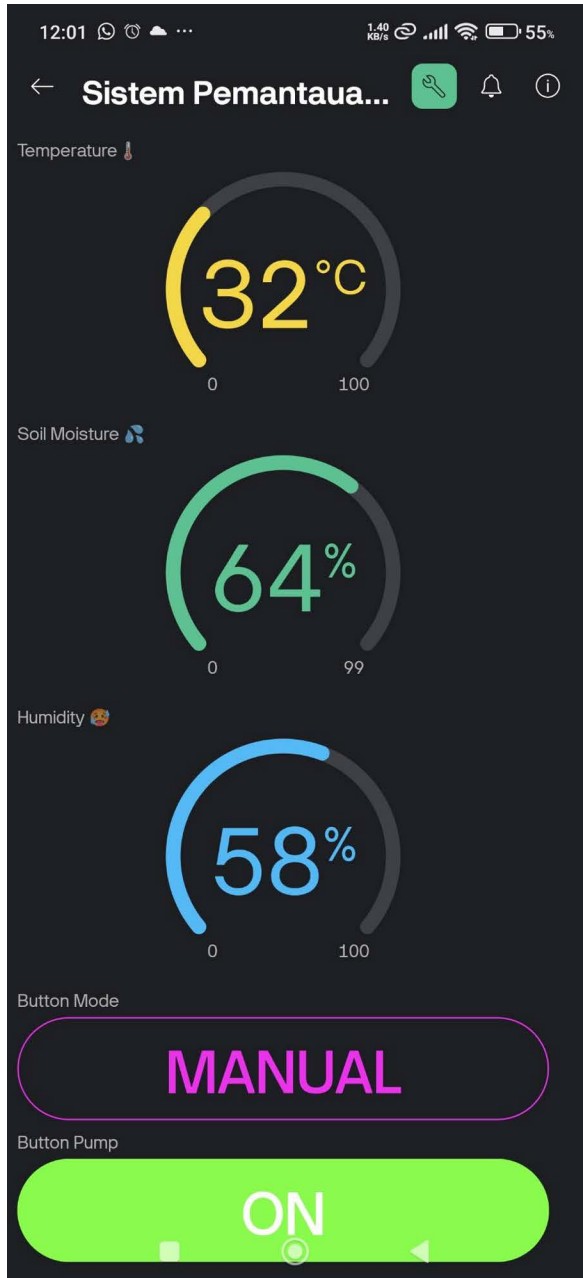

**Fig 8. Blynk IoT mobile app dashboard - Manual control mode.**

Moisture Sensor V2.0, reveals a strong linear relationship between the injected water volume and the increase in soil moisture. For example, applying 10 mL of water resulted in a 29% increase in moisture, while 20 mL increased the moisture to 53%, and 50 mL reached a maximum of 100% moisture. The V2.0 Capacitive Soil Moisture Sensor demonstrated excellent precision and accuracy during testing. Precision, which is calculated based on repeated measurements under consistent conditions, consistently came in at 100%, indicating highly reliable readings. In addition, the accuracy of the sensor reached 100%, ensuring that the measured value was in line with the actual soil moisture level. These

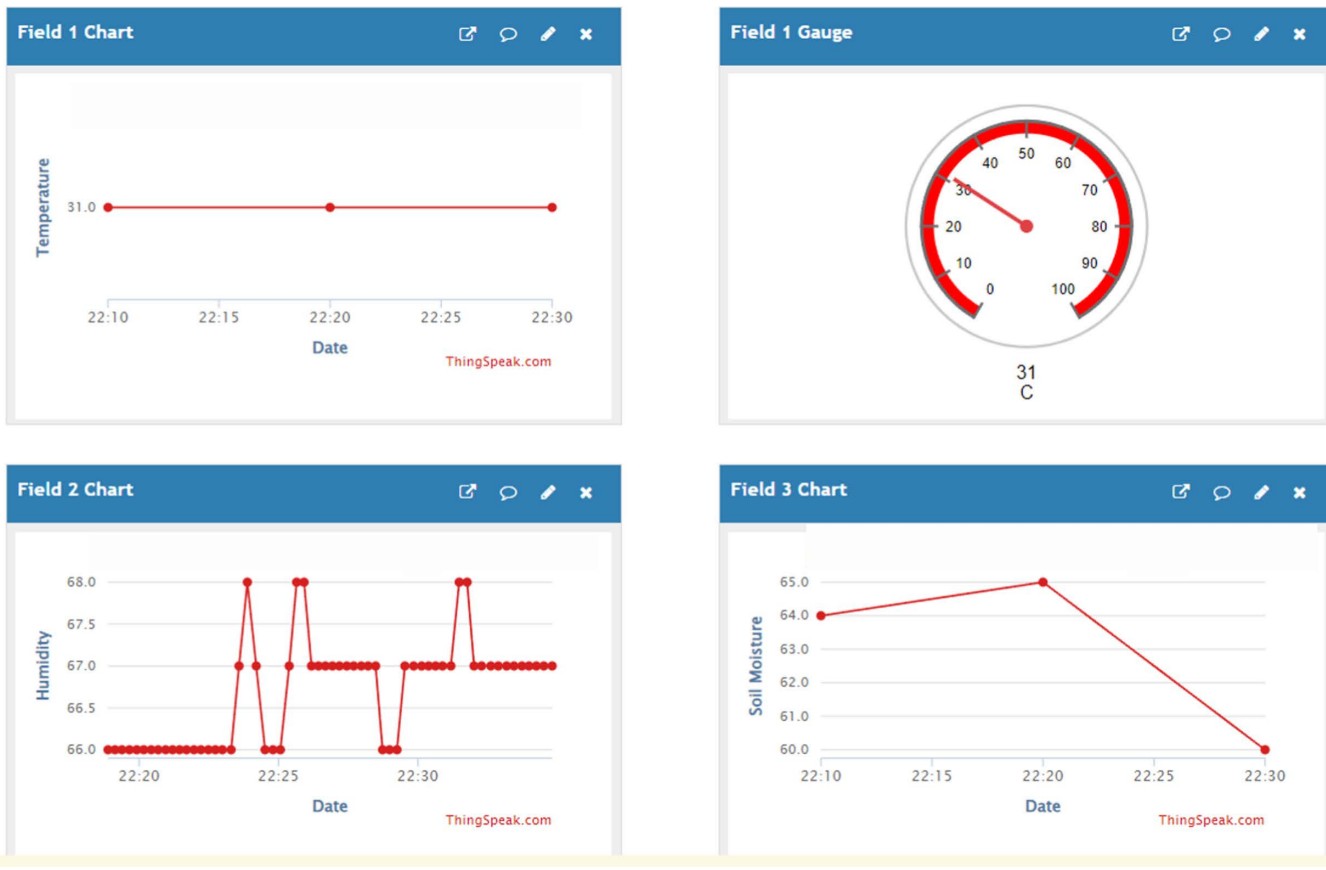

**Fig 9. Thingspeak database website dashboard.**

Table 3. Air temperature measurement: Hygrometer vs. DS18B20 sensor.

| No | Actual Hygrometer Temperature (°C) | DS18B20 Sensor Air Temperature Results (°C) | | | | | Mean (°C) | STD | Bias (°C) | Precision (%) | Accuracy (%) | Error (%) |
|---|---|---|---|---|---|---|---|---|---|---|---|---|
| | | 1 | 2 | 3 | 4 | 5 | | | | | | |
| 1 | 31 | 31 | 31 | 32 | 32 | 32 | 31.6 | 0.5 | -0.6 | 98.45 | 97.19 | 2.81 |
| 2 | 38.2 | 38 | 39 | 39 | 39 | 40 | 39.0 | 0.6 | -0.8 | 98.38 | 97.13 | 2.87 |
| 3 | 42 | 42 | 43 | 43 | 44 | 44 | 43.2 | 0.7 | -1.2 | 98.27 | 97.51 | 2.49 |
| 4 | 45 | 44 | 45 | 46 | 46 | 46 | 45.4 | 0.8 | -0.4 | 98.24 | 95.56 | 4.44 |
| 5 | 48.3 | 48 | 49 | 49 | 50 | 50 | 49.2 | 0.7 | -0.9 | 98.48 | 97.22 | 2.78 |
| Average | | | | | | | | | -0.8 | 98.4 | 96.9 | 3.1 |

results confirm the suitability of the sensor for precise monitoring in automated irrigation systems.

The strong linear relationship between water volume and soil moisture is evident from the data in Table 7, where an increase in the injected water volume consistently leads to a corresponding increase in soil moisture, measured by the Capacitive Soil Moisture Sensor V2.0. This relationship demonstrates the predictive and direct effect that the volume of water has on the moisture content of the soil. For instance, 10 mL of water results in a 29% increase in moisture, while 50 mL achieves a maximum of 100% moisture, showing that the system can reliably measure and react to changes in moisture levels.

**Table 4. Air temperature measurement after calibration: Hygrometer vs. DS18B20 sensor.**

| No | Actual Hygrometer Temperature (°C) | DS18B20 Sensor Air Temperature Results (°C) | | | | | Mean (°C) | STD | Bias (°C) | Precision (%) | Accuracy (%) | Error (%) |
|---|---|---|---|---|---|---|---|---|---|---|---|---|
| | | 1 | 2 | 3 | 4 | 5 | | | | | | |
| 1 | 30.7 | 30 | 30 | 30 | 30 | 30 | 30.0 | 0.0 | 1 | 30.7 | 30 | 30 |
| 2 | 30.7 | 30 | 30 | 30 | 30 | 30 | 30.0 | 0.0 | 2 | 30.7 | 30 | 30 |
| 3 | 30.7 | 30 | 30 | 30 | 30 | 30 | 30.0 | 0.0 | 3 | 30.7 | 30 | 30 |
| 4 | 30.7 | 30 | 30 | 30 | 30 | 30 | 30.0 | 0.0 | 4 | 30.7 | 30 | 30 |
| 5 | 30.5 | 30 | 30 | 30 | 30 | 30 | 30.0 | 0.0 | 5 | 30.5 | 30 | 30 |
| Average | | | | | | | | | | | | |

**Table 5. Air humidity measurement: Hygrometer vs. DHT11 sensor.**

| No | Actual Hygrometer Humidity (%) | DHT11 Sensor Air Humidity Results (%) | | | | | Mean (%) | STD | Bias (%) | Precision (%) | Accuracy (%) | Error (%) |
|---|---|---|---|---|---|---|---|---|---|---|---|---|
| | | 1 | 2 | 3 | 4 | 5 | | | | | | |
| 1 | 61 | 60 | 61 | 61 | 62 | 62 | 61.2 | 0.75 | −0.2 | 98.78 | 96.65 | 3.35 |
| 2 | 60 | 59 | 60 | 61 | 61 | 61 | 60.4 | 0.80 | −0.4 | 98.68 | 96.67 | 3.33 |
| 3 | 60 | 59 | 59 | 60 | 60 | 61 | 59.8 | 0.75 | 0.2 | 98.75 | 95.93 | 4.07 |
| 4 | 60 | 60 | 60 | 61 | 61 | 61 | 60.6 | 0.49 | −0.6 | 99.19 | 98.55 | 1.45 |
| 5 | 59 | 58 | 58 | 59 | 59 | 60 | 58.8 | 0.75 | 0.2 | 98.73 | 95.86 | 4.14 |
| Average | | | | | | | | | −0.2 | 98.8 | 96.7 | 3.3 |

**Table 6. Air humidity measurement: Hygrometer vs. DHT11 sensor after calibration.**

| No | Actual Hygrometer Humidity (%) | DHT11 Sensor Air Humidity Results (%) | | | | | Mean (%) | STD | Bias (%) | Precision (%) | Accuracy (%) | Error (%) |
|---|---|---|---|---|---|---|---|---|---|---|---|---|
| | | 1 | 2 | 3 | 4 | 5 | | | | | | |
| 1 | 64 | 65 | 65 | 65 | 65 | 65 | 65.0 | 0.00 | 1 | 64 | 65 | 65 |
| 2 | 59 | 58 | 58 | 58 | 58 | 58 | 58.0 | 0.00 | 2 | 59 | 58 | 58 |
| 3 | 58 | 58 | 58 | 58 | 58 | 58 | 58.0 | 0.00 | 3 | 58 | 58 | 58 |
| 4 | 56 | 56 | 56 | 56 | 56 | 56 | 56.0 | 0.00 | 4 | 56 | 56 | 56 |
| 5 | 51 | 51 | 51 | 51 | 51 | 51 | 51.0 | 0.00 | 5 | 51 | 51 | 51 |
| Average | | | | | | | | | 0.0 | 100.0 | 100.0 | 0.0 |

**Table 7. Soil moisture (%) increase in response to water volume.**

| No | Water Volume (mL) | Capacitive Soil Moisture Sensor V2.0 Results (%) | | | | | Mean (%) | STD | Bias (%) | Precision (%) | Accuracy (%) | Error (%) |
|---|---|---|---|---|---|---|---|---|---|---|---|---|
| | | 1 | 2 | 3 | 4 | 5 | | | | | | |
| 1 | 10 | 29 | 29 | 29 | 29 | 29 | 29.0 | 0.00 | −19.0 | 100.00 | 100.00 | 0.00 |
| 2 | 20 | 53 | 53 | 53 | 53 | 53 | 53.0 | 0.00 | −33.0 | 100.00 | 100.00 | 0.00 |
| 3 | 30 | 72 | 72 | 72 | 72 | 72 | 72.0 | 0.00 | −42.0 | 100.00 | 100.00 | 0.00 |
| 4 | 40 | 82 | 82 | 82 | 82 | 82 | 82.0 | 0.00 | −42.0 | 100.00 | 100.00 | 0.00 |
| 5 | 50 | 100 | 100 | 100 | 100 | 100 | 100.0 | 0.00 | −50.0 | 100.00 | 100.00 | 0.00 |
| Average | | | | | | | | | −37.2 | 100.0 | 100.0 | 0.0 |

However, the data also suggests the presence of threshold effects at higher water volumes. As the water volume increases beyond 40 mL, the increase in soil moisture becomes less pronounced, reaching 100% moisture with 50 mL. This diminishing return implies that there is a point at which adding more water does not significantly improve soil moisture content. Once the soil reaches saturation, additional water may not be effectively absorbed, potentially leading to waterlogging, which can be detrimental to plant health by reducing root oxygenation and promoting anaerobic conditions.

### 3.5 Feasibility and broader implications of IoT-based automatic irrigation systems

A cost analysis was conducted to assess the feasibility of implementing an IoT-based automatic irrigation system for smallholder farmers. This system, built using a NodeMCU microcontroller, DS18B20 temperature sensor, DHT11 humidity sensor, soil moisture sensor, relay module, DC pump, LCD display, and additional peripherals, has a hardware cost of IDR 235,000 (~USD 15.00). The system operates on a 5V USB power source with an average power consumption of 1.0W, resulting in minimal operational costs. Daily energy consumption is 24Wh or 0.024kWh, calculated as 1.0W × 24 hours. Based on an electricity rate of IDR 1,352/kWh, the daily electricity cost amounts to IDR 32.45. Over a year, this translates to an annual electricity cost of approximately IDR 11,843.52 with a calculation of IDR 32.45/day × 365 days.

When compared to traditional irrigation, which requires 1000mL of water per plant, twice a day, this IoT-based system reduces water usage by 30% through real-time monitoring and optimization of irrigation based on soil moisture levels. By maintaining soil moisture within the ideal range of 30%–60%, the system prevents overwatering and minimizes water wastage, promoting healthier plant growth. The soil moisture levels are categorized as dry (0%–30%), moist (30%–60%), and wet (60%–100%), and the system continuously monitors these conditions using sensors to ensure optimal irrigation.

This IoT-based solution is particularly impactful for smallholder farmers, offering a scalable and cost-effective approach to sustainable water management. The modular design allows for customization to suit varying farm sizes and types, making it adaptable for subsistence farming or small-scale agricultural operations. By reducing water usage and increasing irrigation efficiency, the system promotes sustainability while supporting improved crop productivity.

For larger agricultural operations, the benefits are equally compelling. The system's ability to automate irrigation processes and provide real-time monitoring across expansive areas optimizes water use and minimizes wastage. This level of precision can significantly enhance crop yields while reducing labor and operational costs associated with traditional irrigation methods. Additionally, continuous data collection and analysis enable informed decision-making, further improving farm management efficiency and sustainability.

The broader implications of adopting IoT-based irrigation systems are transformative. By making advanced technology accessible and adaptable for farmers of all scales, these systems encourage the widespread adoption of sustainable practices. The resulting improvements in water efficiency and crop productivity contribute to long-term agricultural resilience, fostering food security and environmental sustainability across diverse farming contexts. This aligns directly with the United Nations Sustainable Development Goals (SDGs), specifically SDG 2: Zero Hunger, SDG 6: Clean Water and Sanitation, and SDG 13: Climate Action. By addressing these goals, IoT-based irrigation systems support the creation of resilient agricultural systems that can adapt to climate change, ensure efficient resource use, and provide equitable access to technological advancements for all farmers.

## 4 Conclusions

This study successfully developed an IoT-based automatic irrigation system that addresses the challenge of water scarcity in sustainable agriculture, particularly in regions with limited rainfall. The system utilizes the Blynk platform and ThingSpeak database for real-time monitoring and precise water control, integrating a DS18B20 temperature sensor, DHT11 humidity sensor, and soil moisture sensor through an ESP8266 NodeMCU module for seamless connectivity.

Testing results revealed that the DS18B20 sensor achieved 98.4% precision and 96.9% accuracy, with a bias of −0.9°C, while the DHT11 sensor showed 98.8% precision, 96.7% accuracy, a −0.2% average bias, and a 3.3% error margin. After calibration, both sensors reached 100% precision and accuracy, closely aligning with reference standards. Soil moisture data indicated a strong correlation between irrigation volume and moisture levels, where an additional 10 mL of water increased soil moisture by 29%.

This IoT-enabled system provides farmers with enhanced control over irrigation, making it possible to optimize water use and make data-driven decisions. The findings validate the system's potential to improve water management in agriculture, supporting sustainable practices with high precision and accuracy. Future improvements could include integrating machine learning for predictive analytics to further enhance irrigation efficiency by anticipating water needs based on weather patterns and soil conditions. Additionally, expanding the system to include other environmental sensors or integrating it with broader smart farming technologies could offer a solution for sustainable agricultural management.

## Supporting information

**S1 File. Supporting information.**
(DOCX)

## Acknowledgments

The authors would like to sincerely acknowledge Universitas Multimedia Nusantara (UMN) for facilities support in this research. Their support has been instrumental in the successful completion of this study. The authors are deeply grateful for the opportunity to conduct this research with their assistance, which has significantly contributed to its development and outcomes.

## Author contributions

**Conceptualization:** Fahmy Rinanda Saputri.

**Funding acquisition:** Fahmy Rinanda Saputri.

**Investigation:** Ricardo Linelson.

**Methodology:** Fahmy Rinanda Saputri.

**Project administration:** Muhammad Salehuddin.

**Software:** Ricardo Linelson.

**Supervision:** Fahmy Rinanda Saputri.

**Validation:** Fahmy Rinanda Saputri, Muhammad Salehuddin, Danial Md Nor, Muhammad Imran Ahmad.

**Visualization:** Ricardo Linelson.

**Writing – original draft:** Ricardo Linelson.

**Writing – review & editing:** Fahmy Rinanda Saputri.

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
