## [Decision Letter · Decision Letter 0]

21 Jan 2025

PONE-D-24-54867Design and development of an irrigation monitoring and control system based on blynk internet of things and thingspeakPLOS ONE

Dear Dr. Saputri,

Thank you for submitting your manuscript to PLOS ONE. After careful consideration, we feel that it has merit but does not fully meet PLOS ONE’s publication criteria as it currently stands. Therefore, we invite you to submit a revised version of the manuscript that addresses the points raised during the review process. **Comments on the manuscript**

Dear Authors, I appreciate your efforts in this work. However, the paper requires significant improvement before it can be considered for publication. The current submission leans more toward a traditional IoT application rather than presenting novel research. Below are my comments and recommendations to enhance the quality of your manuscript:

In the abstract, (i) clearly state the primary aim or purpose of the research, specifying the exact problem or gap being addressed, and (ii) Highlight the uniqueness or innovation of your work to justify its contribution.Also, in the abstract, mentioning the significant results should be brief, leaving detailed explanations for the relevant sections of the paper.The introduction and appended literature review are limited and require substantial improvement. (i) Provide a comprehensive background that contextualizes your research, supported by recent and relevant references. (ii) Include a robust literature review to highlight existing works, their advantages and limitations, and the specific gap your research aims to address.Avoid unnecessary repetition across sections, particularly where you have described your proposed system.Clearly state the unique features of your system compared to existing solutions. If your work is merely a classical application of IoT, it may lack the novelty required for publication.Section 2.1 "Research Approach" reads as a system description rather than a research approach. Revise the section title and content to align with its purpose or provide justification for the current format.In Section 2.2, you state that the research began with a literature review. However, no substantial literature review is presented in the manuscript. Address this discrepancy by including a proper review as noted in point 2.Include a block diagram of your system before presenting the schematic diagram.Improve the readability of labels in Figure 1 ("Schematic Diagram").Correct the captions for Figures 2 and 4, as they currently have identical captions.Restructure the logical flow of your sections. For example, the content in Section 2.3 (“Research Procedure and Algorithm”) on page 14 should appear before Section 2.2 (“Research Design”), as it describes the research procedure. however, some of the content on page 15 could remain after Section 2.2.Configuration parameters and other settings in Section 2.2 should be summarized in a table to improve readability.Provide details about the deployment environment, location, and context for the prototype testing and validation.In addition to Figure 5 (“Final Circuit Design”), you should include images of the complete functioning prototype in the testing environment, showing all components, including the pump and other components.The "Discussion" section is misleading, as it primarily presents results rather than providing analysis or interpretation. Move all results, including tables and relevant text, to the "Results" section. Then, revise the "Discussion" section to focus on analyzing and interpreting your findings, explaining the reasons behind the results, and linking them to the research objectives and existing literature to emphasize their significance

In conclusion, these required improvements are essential to elevate this work to a research-level contribution. I hope these recommendations, along with reviewer comments, as well as your expertise, help you strengthen the manuscript for resubmission in the second review round.

We look forward to receiving your revised manuscript.

Kind regards,

Antar S. H. Abdul-Qawy, Ph.D.

Academic Editor

PLOS ONE

“The authors would like to sincerely acknowledge Universitas Multimedia Nusantara (UMN) for funding this research through the grant (Number 0007-RD-LPPM-UMN/P-INT/VI/2024). Their financial support has been instrumental in the successful completion of this study. The authors are deeply grateful for the opportunity to conduct this research with their assistance, which has significantly contributed to its development and outcomes.”

5. We note that you have indicated that there are restrictions to data sharing for this study. PLOS only allows data to be available upon request if there are legal or ethical restrictions on sharing data publicly. For more information on unacceptable data access restrictions, please see http://journals.plos.org/plosone/s/data-availability#loc-unacceptable-data-access-restrictions.

Additional Editor Comments:

**Important Note: **

Please not that some reviewers have recommended several citations in their review. I recommend that you thoroughly evaluate these references and determine whether they are relevant to the current study or not. Feel free to disregard any references that do not align with the content of the manuscript. This will not affect the acceptance or rejection of your paper.

Comments from the editorial office and Academic Editor: Upon internal evaluation of the reviews provided, we kindly request you to disregard the reviewer report provided by Reviewer 2. No amendments are required in response to reviewer 2’s comments.

Reviewers' comments:

Reviewer's Responses to Questions

**Comments to the Author**

1. Is the manuscript technically sound, and do the data support the conclusions?

Reviewer #1: Yes

Reviewer #2: Yes

Reviewer #3: Yes

Reviewer #4: Yes

2. Has the statistical analysis been performed appropriately and rigorously? 

Reviewer #1: Yes

Reviewer #2: Yes

Reviewer #3: Yes

Reviewer #4: Yes

3. Have the authors made all data underlying the findings in their manuscript fully available?

Reviewer #1: Yes

Reviewer #2: Yes

Reviewer #3: No

Reviewer #4: Yes

4. Is the manuscript presented in an intelligible fashion and written in standard English?

Reviewer #1: Yes

Reviewer #2: Yes

Reviewer #3: No

Reviewer #4: Yes

5. Review Comments to the Author

Reviewer #1: This paper talks about “Design and development of an irrigation monitoring and control system based on blynk internet of things and thingspeak”. While the paper has merit, there is room for improvement in terms of quality. Dear authors, I have several comments and questions. Below are my suggestions for revising the manuscript to enhance its overall quality. I accept this paper after addressing these comments:

1) The problem statement in the abstract is missing. Also, should be added the significance of this study.

2) Keywords should be in alphabetical order.

3) The context and problem statement in the introduction are missing. It should be more elaborated.

4) The introduction writing part is not satisfactory. Need to be improved. The novelty should be highlighted in the introduction.

5) The review of the literature needs more updating with works to have a clear and concise state-of-the-art analysis. This should more clearly show the knowledge gaps identified and link them to the paper's goals.

6) What is the focus of the work? This should be added in the introduction.

7) The manuscript (Introduction) has deficiency citations to very important works published before, you can add these works:

https://doi.org/10.1016/j.rineng.2024.103392

https://doi.org/10.1016/j.rineng.2024.102705

https://doi.org/10.1016/j.rineng.2024.102705

https://doi.org/10.1016/j.jssas.2024.02.001

https://doi.org/10.1016/j.rineng.2024.102829.

https://doi.org/10.1016/j.cles.2024.100132.

https://ieeexplore.ieee.org/document/10548972.

8) What is the novelty of this study? It should be added.

9) Please add the main contribution of this work in the introduction. The organization of this paper should also be added in the last introduction section.

10) Please add more detail in the section on the “Materials and Methods”. Also, pseudocode should be added. This section should be improved.

11) In the section of results, a more detailed description of the results presented in the figures and tables should be provided, incorporating both qualitative and quantitative analyses.

12) Please include a comparative analysis in tabular form, highlighting the findings of this study about other recent studies.

13) The quality of the Figures should be improved.

14) Please include a list of abbreviations.

Reviewer #2: I have reviewed the manuscript titled "Design and Development of an Irrigation Monitoring and Control System Based on Blynk IoT and ThingSpeak." This study proposes an IoT-enabled automatic irrigation system that addresses water scarcity challenges in sustainable agriculture. By integrating Blynk and ThingSpeak, the research advances real-time monitoring and efficient water management practices. Below, I provide constructive feedback to help refine the manuscript further.

Content and Structure

The manuscript is well-organized, with clear sections detailing the objectives, methodology, results, and conclusions. However, several areas could benefit from further refinement:

Abstract:

The abstract is concise and informative but should include key performance metrics (e.g., water efficiency improvements, precision/accuracy values) to quantify the system’s effectiveness.

Highlight the broader implications of this work for scalable IoT-based agricultural practices.

Introduction:

The introduction effectively outlines the challenges of water management in agriculture and the relevance of IoT-based solutions. However, it could expand on:

Existing limitations of conventional irrigation systems and previous IoT approaches.

How this study uniquely addresses these limitations, particularly through the integration of Blynk and ThingSpeak.

Conclusion:

While the conclusion summarizes the findings well, it could:

Include more specific recommendations for future work, such as scalability for larger farms or integration with AI-based predictive analytics.

Address potential challenges, such as maintenance costs or the system's adaptability to diverse climates.

Literature Review and Citations

The literature review provides a strong foundation for the study, citing relevant works on IoT and automated irrigation. To further enrich this section, I suggest including recent studies that explore:

Advanced IoT frameworks for resource optimization.

The role of hybrid sensor networks in precision agriculture.

Suggested references to consider:

https://doi.org/10.1016/j.eswa.2023.122147: Discusses optimization techniques relevant to IoT systems.

https://doi.org/10.54216/JAIM.080103: Focuses on robust machine learning approaches in smart agriculture.

Technical Clarifications and Suggestions

System Architecture:

The description of the system components (DS18B20, DHT11, NodeMCU) is clear, but further detail is needed on:

The calibration process for sensors and how it ensures long-term accuracy.

The rationale for selecting the 60% moisture threshold and its applicability across different soil types.

Data Integration:

Elaborate on how the ThingSpeak database processes and visualizes data. Could alternative platforms or APIs enhance functionality or scalability?

Discuss the computational overhead of real-time data transmission and control.

Energy Efficiency:

Include a discussion on energy consumption, especially for the NodeMCU and pump operation. Could renewable energy sources (e.g., solar panels) be integrated for off-grid farming?

Scalability:

The current design is tailored for small-scale farms. Discuss the potential modifications needed for deployment in large-scale agricultural operations.

Visualization and Analysis

The manuscript incorporates useful visualizations, but additional enhancements could improve clarity and impact:

Circuit Design (Figure 5):

Add annotations or labels to highlight key components and their interconnections for readers unfamiliar with hardware schematics.

Data Visualization:

The ThingSpeak dashboard (Figure 7) effectively visualizes data trends but could include:

Comparative plots showing pre- and post-calibration sensor readings.

Water usage trends over extended periods to quantify efficiency improvements.

Performance Metrics:

Summarize the bias, precision, and accuracy values for all sensors in a consolidated table for quick reference.

Practical Applications

The manuscript outlines potential benefits but could expand on real-world applications and challenges:

Implementation in Developing Regions:

Discuss how this system addresses specific barriers in resource-limited settings, such as affordability, connectivity issues, or lack of technical expertise.

Policy Implications:

Highlight how this work aligns with Sustainable Development Goals (SDGs), particularly those related to water conservation and food security.

Integration with Predictive Analytics:

Explore opportunities to integrate predictive algorithms for rainfall patterns or crop-specific irrigation schedules.

Future Directions

The manuscript briefly mentions future improvements but should provide a more detailed roadmap:

Enhanced Sensing Capabilities:

Incorporate additional environmental sensors (e.g., pH, salinity) to broaden the system’s applicability.

AI and Machine Learning:

Investigate the use of AI for adaptive irrigation based on historical and real-time data.

Cost Optimization:

Explore ways to reduce hardware costs while maintaining system reliability.

Summary

This manuscript makes a valuable contribution to IoT-based sustainable agriculture by presenting a practical, low-cost solution for automated irrigation. The proposed system is innovative and well-documented, with significant potential for real-world applications. By addressing the suggested revisions, the study can make an even greater impact on precision agriculture and water resource management.

Reviewer #3: The paper proposes the design and development of an IoT-based automatic irrigation monitoring and control system. The system integrates real-time data collection and control using sensors (temperature, humidity, and soil moisture), the Blynk IoT platform for monitoring and control, and the ThingSpeak database for historical data storage and analysis. The aim is to optimize water usage and support sustainable agriculture practices, addressing water scarcity exacerbated by climate change. The main novelty is the integration of Blynk for real-time control and ThingSpeak for data storage—bridging previous gaps in IoT-based irrigation systems. Il also uses DS18B20 sensors and calibration improvements to enhance sensor precision and accuracy. The solution is low-cost and scalable tailored for resource-constrained settings.

My concerns are:

1. How do the precision and accuracy metrics relate to the practical benefits for end users, and could they be improved to highlight real-world impact?

2. What specific comparative advantages does the system offer over existing systems, and how can these be emphasized more clearly?

3. To what extent has the economic feasibility and scalability of the system for large farms been explored, and what insights can be gained in this area?

4. How clearly is the problem statement defined in terms of user needs and regional variability, and what aspects need further clarification?

5. Does the methodology account for potential variability in sensor performance under extreme conditions, and how might this affect results?

6. Can the calibration process be explained in more detail beyond numerical results, especially in terms of real-world implementation?

7. Why does the experimentation focus on technical metrics (accuracy, precision, error) without field validation in diverse environmental or geographic conditions?

8. How does the system’s reliance on Wi-Fi connectivity limit its applicability in remote agricultural areas, and what alternatives could be explored?

9. What are the reasons for the absence of integration with predictive models for weather or crop-specific requirements, and how could this enhance the system?

10. How effectively does the system adapt to large-scale agricultural operations, and are there challenges in scaling it up?

11. How should long-term maintenance requirements and costs associated with sensors and cloud infrastructure be addressed?

12. Are there sufficient benchmarks or comparative studies with existing IoT-based irrigation solutions, and how could these be expanded upon?

13. How reliant is the system on reliable internet connectivity for real-time monitoring, and what impact does this have in areas with limited access?

14. Why is there no inclusion of energy efficiency analysis for the system’s long-term operation, and how might this be addressed?

15. Has the exploration of alternative power sources, such as solar energy for remote installations, been considered in the system design?

16. What is the rationale behind the lack of multi-year deployment studies to assess the durability and real-world performance of the system?

17. How can field tests be extended to varied geographic and climatic regions to establish the system’s robustness?

18. Could an economic analysis be incorporated to assess the cost-benefit ratio for farmers and highlight its viability?

19. How can predictive algorithms for weather-based irrigation adjustments be integrated into the system?

20. Would exploring alternative communication technologies (e.g., LoRa) help in areas with poor internet access, and what would the implications be?

Reviewer #4: The manuscript discusses an IoT-based automatic irrigation system, demonstrating its potential in sustainable agriculture. The work is promising, but several areas need refinement to enhance clarity, technical rigor, and overall readability. I have included my comments to the attached document

6. PLOS authors have the option to publish the peer review history of their article (what does this mean? ). If published, this will include your full peer review and any attached files.

**Do you want your identity to be public for this peer review?** For information about this choice, including consent withdrawal, please see our Privacy Policy .

Reviewer #1: No

Reviewer #2: No

Reviewer #3: No

Reviewer #4: **Yes: ** Ibrahim L. Kadigi

---

## [Author Response · Author response to Decision Letter 1]

29 Jan 2025

Dear Reviewers and Editors,

The authors of this paper greatly appreciate the valuable comments from the reviewers and editors. The suggestions are greatly helpful in improving the quality of our paper. We have made careful modifications strictly according to the reviewers' advice.

Our detailed responses to the reviewers' comments, along with the revised manuscript, have been uploaded as accompanying files for your review.

Sincerely,

Authors

---

## [Editor Report · Decision Letter 1]

3 Feb 2025

PONE-D-24-54867R1Design and development of an irrigation monitoring and control system based on blynk internet of things and thingspeakPLOS ONE

Dear Dr. Saputri,

Thank you for submitting your manuscript to PLOS ONE. After careful consideration, we feel that it has merit but does not fully meet PLOS ONE’s publication criteria as it currently stands. Therefore, we invite you to submit a revised version of the manuscript that addresses the points raised during the review process.

Dear Authors,

I have assessed the revised manuscript along with your responses to the reviewers' concerns. I determined that they have not been completely addressed. While you responded to Reviewer #4’s comments, the comments from Reviewer #1, Reviewer #3, and my own comments remain completely unaddressed.

Therefore, I am issuing another revision decision and requiring you to address the following:

**1. Editor’s Comments**

**2. Reviewer #1’s Comments**

**3. Reviewer #3’s Comments**

All these comments and details were provided in the first-round decision letter. Please revisit that letter carefully and respond accordingly.

**Please note:** If these concerns remain unaddressed in the next revision, the manuscript may be rejected.

We look forward to receiving your revised manuscript.

Kind regards,

Antar S. H. Abdul-Qawy, Ph.D.

Academic Editor

PLOS ONE

---

## [Author Response · Author response to Decision Letter 2]

5 Feb 2025

Dear Dr. Abdul-Qawy,

Thank you for your detailed feedback on our manuscript titled "Design and Development of an Irrigation Monitoring and Control System Based on Blynk Internet of Things and ThingSpeak". We appreciate the opportunity to revise our work.

We have carefully addressed the comments from Reviewer #1, Reviewer #2, Reviewer #3, and yourself. The revised manuscript has been updated accordingly, and we have provided detailed responses to each of the reviewer comments in a separate document titled "Response to Reviewers".

Additionally, we have uploaded the following files as part of our revision submission:

Revised Manuscript – with all changes clearly tracked in the document ("Revised Manuscript with Track Changes").

Clean Manuscript – the final version without track changes ("Manuscript").

Response to Reviewers – a document detailing how we addressed each comment.

We hope the revisions meet the journal’s expectations and look forward to hearing your feedback.

Please let us know if you require any further clarifications or additional revisions.

Kind regards,

Fahmy Rinanda Saputri

---

## [Decision Letter · Decision Letter 2]

4 Mar 2025

Design and development of an irrigation monitoring and control system based on blynk internet of things and thingspeak

PONE-D-24-54867R2

Dear Dr. Saputri,

We’re pleased to inform you that your manuscript has been judged scientifically suitable for publication and will be formally accepted for publication once it meets all outstanding technical requirements.

Kind regards,

Antar S. H. Abdul-Qawy, Ph.D.

Academic Editor

PLOS ONE

Additional Editor Comments (optional):

Dear Author, please enhance the clarity of the labels in Figures 1, 2, 3, 4, and 7

Reviewers' comments:

Reviewer's Responses to Questions

**Comments to the Author**

1. If the authors have adequately addressed your comments raised in a previous round of review and you feel that this manuscript is now acceptable for publication, you may indicate that here to bypass the “Comments to the Author” section, enter your conflict of interest statement in the “Confidential to Editor” section, and submit your "Accept" recommendation.

Reviewer #1: All comments have been addressed

2. Is the manuscript technically sound, and do the data support the conclusions?

Reviewer #1: Yes

3. Has the statistical analysis been performed appropriately and rigorously? 

Reviewer #1: Yes

4. Have the authors made all data underlying the findings in their manuscript fully available?

Reviewer #1: Yes

5. Is the manuscript presented in an intelligible fashion and written in standard English?

Reviewer #1: Yes

6. Review Comments to the Author

Reviewer #1: The authors have addressed the comments, and the paper has been improved. Therefore, it is recommended to accept the publication of this paper in its current form.

7. PLOS authors have the option to publish the peer review history of their article (what does this mean? ). If published, this will include your full peer review and any attached files.

**Do you want your identity to be public for this peer review?** For information about this choice, including consent withdrawal, please see our Privacy Policy .

Reviewer #1: No

---

## [Editor Report · Acceptance letter]

PONE-D-24-54867R2

PLOS ONE

Dear Dr. Saputri,

I'm pleased to inform you that your manuscript has been deemed suitable for publication in PLOS ONE. Congratulations! Your manuscript is now being handed over to our production team.

Kind regards,

on behalf of

Dr. Antar S. H. Abdul-Qawy

Academic Editor

PLOS ONE